# Thebromine Targets Adenosine Receptors to Control Hippocampal Neuronal Function and Damage

**DOI:** 10.3390/ijms231810510

**Published:** 2022-09-10

**Authors:** Pedro Valada, Sofia Alçada-Morais, Rodrigo A. Cunha, João Pedro Lopes

**Affiliations:** 1CNC-Center for Neuroscience and Cell Biology, University of Coimbra, 3004-504 Coimbra, Portugal; 2Faculty of Medicine, University of Coimbra, 3004-504 Coimbra, Portugal

**Keywords:** theobromine, caffeine, adenosine receptors, synaptic transmission, synaptic plasticity, Alzheimer’s disease

## Abstract

Theobromine is a caffeine metabolite most abundant in dark chocolate, of which consumption is linked with a lower risk of cognitive decline. However, the mechanisms through which theobromine affects neuronal function remain ill-defined. Using electrophysiological recordings in mouse hippocampal synapses, we now characterized the impact of a realistic concentration of theobromine on synaptic transmission and plasticity. Theobromine (30 μM) facilitated synaptic transmission while decreasing the magnitude of long-term potentiation (LTP), with both effects being blunted by adenosine deaminase (2 U/mL). The pharmacological blockade of A_1_R with DPCPX (100 nM) eliminated the theobromine-dependent facilitation of synaptic transmission, whereas the A_2A_R antagonist SCH58261 (50 nM), as well as the genetic deletion of A_2A_R, abrogated the theobromine-induced impairment of LTP. Furthermore, theobromine prevented LTP deficits and neuronal loss, respectively, in mouse hippocampal slices and neuronal cultures exposed to Aβ_1–42_ peptides, considered a culprit of Alzheimer’s disease. Overall, these results indicate that theobromine affects information flow via the antagonism of adenosine receptors, normalizing synaptic plasticity and affording neuroprotection in dementia-related conditions in a manner similar to caffeine.

## 1. Introduction

Caffeine is the most widely consumed psychoactive drug, enhancing well-being, alertness and attention [1]. Epidemiological studies have also shown that the regular consumption of a moderate amount of coffee (3–5 cups per day) increases memory consolidation [2,3] and decreases cognitive decline and, in particular, the risk of developing AD (e.g., [4,5,6]). Accordingly, animal studies indicate that caffeine triggers pro-cognitive adaptive changes in the brain [7] and attenuates memory deficits in the animal models of different brain disorders (reviewed in [8]), including AD (e.g., [9,10]). Caffeine modulates information flow in neuronal circuits, enhancing basal synaptic transmission through the antagonism of adenosine A_1_ receptors (A_1_R) [11,12] and dampens hippocampal long-term potentiation (LTP) [11,13], a form of synaptic plasticity considered the neurophysiological basis of memory [14], through the antagonism of adenosine A_2A_ receptors (A_2A_R). The neuroprotective effect of caffeine in different animal models of neuropsychiatric disorders, including AD [9,10], has been linked to the ability of caffeine to normalize synaptic plasticity via A_2A_R (reviewed in [15]), since the overactivation of hippocampal A_2A_R is sufficient to trigger memory impairment [16,17,18] and is critically necessary for the emergence of synaptic and memory deficits in different animal models of early AD [19,20,21,22].

In a recent attempt to correlate the levels of caffeine with the currently established neurochemical markers of AD, we were surprised to conclude that it was the levels of theobromine rather than caffeine that correlated inversely with the altered cerebrospinal fluid levels of Aβ_1–42_ and modified tau proteins [23]. This was particularly surprising, since the 1-N-demethylation of caffeine into theobromine only accounts for approximately 7 to 8% of caffeine metabolism, although its liver metabolism through CYP1A2 and CYP2E1 is slower than that of caffeine in humans [24] and different in different mammals [25], making it a toxic compound for some animals (e.g., [26,27]). The association between theobromine and AD was further bolstered by a recent study [28], showing that the aging-associated cognitive decline was attenuated in consumers of chocolate [29], whose main psycho-pharmacologically active constituent is theobromine. Accordingly, in animal models, theobromine has been associated with memory improvement [30,31].

The pharmacological characterization of the targets of xanthines prompted the conclusion that theobromine has a low potency (in the hundreds of micromolar range) to antagonize A_1_R and A_2A_R, compared to caffeine [32,33,34], making it a lesser homologue of caffeine. This has contributed to the poor interest dedicated to possible central effects of theobromine, which is instead clinically used as a vasodilator, a diuretic, and heart stimulant, with renewed interest in tooth decay prevention [35] and as an antitussive [36], possibly due to the particular ability of theobromine to interfere with targets such as poly(ADP-ribose)polymerase-1 [37] or sirtuin-1 [38].

This apparently different pharmacokinetic and pharmacodynamic profile of caffeine and theobromine prompts the need for a clarification of their mechanisms of action in the nervous system to grasp this surprising inverse association between cognitive deterioration and chocolate consumption [28] or theobromine levels [23]. Thus, we now hypothesized and tested if theobromine also acted on neuronal function and viability through the antagonism of adenosine receptors, as previously defined for caffeine [11].

## 2. Results

### 2.1. Theobromine Bolsters Basal Synaptic Transmission and Impairs Long-Term Potentiation

Theobromine enhances basal synaptic transmission in mouse hippocampal slices in a concentration-dependent manner, reaching a plateau after 30 μM (Figure 1A). This is within the range of concentrations of theobromine found in the cerebral cortex of rats that consumed theobromine-supplemented chow for 40 days (1.94 ± 0.16 μg/mL or 10.77 ± 0.89 μM) [39]. Exposure to theobromine (30 μM) increased basal synaptic transmission by 34.04 ± 2.80% (*p* < 0.001 vs. baseline (0%), n = 22), reaching a plateau after approximately 10 min, which remained stable up to 30 min; upon washout, synaptic transmission returned to near-baseline values after 30 min (2.13 ± 2.77%; *p* > 0.05 vs. baseline, n = 22, one sample *t* test) (Figure 1B,C).

Contrariwise, mouse hippocampal slices superfused with 30 μM theobromine displayed an impairment of synaptic plasticity with a significant decrease in long-term potentiation (LTP), with a magnitude reaching 66.99 ± 10.29% in the absence of theobromine vs. 24.39 ± 7.03% in its presence (*p* < 0.01; n = 6; Figure 1D,E). Remarkably, these effects of theobromine on synaptic transmission and plasticity are analogous to those recorded upon the exposure of mouse hippocampal slices to an identical concentration of caffeine [11].

### 2.2. The Effect of Theobromine on Synaptic Transmission and Plasticity Is Lost upon Removal of Extracellular Adenosine

In order to assess if theobromine relies on adenosine neuromodulation to alter synaptic transmission and plasticity, as previously shown for caffeine [11], we probed for the impact of adenosine deaminase (ADA), an enzyme that converts adenosine to inosine, on the synaptic changes elicited by theobromine (30 μM). Clearing endogenous extracellular adenosine increased basal synaptic transmission (32.30 ± 3.31%, *p* < 0.001 vs. baseline, n = 5) and occluded the effect of theobromine on basal synaptic transmission (32.30 ± 3.31% in the presence of ADA vs. 35.10 ± 4.74% in the presence of ADA and theobromine; *p* > 0.05, n = 5; Figure 2A,B). Likewise, ADA also hindered the decrease of LTP magnitude induced by theobromine (23.51 ± 5.09% in the presence of ADA vs. 28.57 ± 10.21% in the presence of ADA and theobromine; *p* > 0.05, n = 4–5; Figure 2C,D).

### 2.3. Theobromine Bolsters Synaptic Transmission through the Antagonism of A_1_R

We next determined the contribution of the two main adenosine receptor subtypes (A_1_R and A_2A_R) to the effects of theobromine on basal synaptic transmission and synaptic plasticity. To detail the involvement of A_1_R we used the selective A_1_R antagonist DPCPX (100 nM). DPCPX increased basal synaptic transmission by 39.91 ± 4.59% (*p* < 0.001 vs. baseline, n = 11; Figure 3A,B) but had no significant effect on LTP (54.16 ± 8.12% in the absence and 38.47 ± 5.44% in the presence of DPCPX; *p* > 0.05, n = 5–6; Figure 3C,D), which is in line with the modest impact of A_1_R antagonism on synaptic plasticity in adult rodents [11,40]. Analogously to what we previously showed for caffeine [11], DPCPX occluded the effect of theobromine (30 μM) on basal synaptic transmission (2.87 ± 8.23% vs. the baseline in the presence of DPCPX; *p* > 0.05, n = 11; Figure 3A,B). However, the antagonism of A_1_R did not alter the impact of theobromine on LTP magnitude (38.47 ± 5.44% in the presence of DPCPX vs. 12.94 ± 4.23% in the presence of DPCPX + theobromine, *p* < 0.05, n = 4–5; Figure 3C,D).

### 2.4. Theobromine Impairs Synaptic Plasticity through the Antagonism of A_2A_R

To define the importance of A_2A_R in the effects of theobromine on synaptic transmission and plasticity, we first used the selective A_2A_R antagonist SCH58261 (50 nM) [40]. SCH58261 did not alter basal synaptic transmission (0.35 ± 1.29% vs. baseline, *p* > 0.05, n = 4) and did not affect the increase in basal synaptic transmission triggered by theobromine (30 μM) (22.74 ± 6.41% vs. baseline in the presence of SCH58261; *p* < 0.05, n = 4) (Figure 4A,B).

Since A_2A_R are selectively involved in the control of hippocampal synaptic plasticity [40,41], we next tested if SCH58261 prevented the effects of theobromine on hippocampal LTP. In agreement with previous observations [11,20,40], SCH58261 (50 nM) significantly decreased LTP magnitude (13.31 ± 3.74% with SCH58261 vs. 70.21 ± 2.36% without SCH58261, *p* < 0.001, n = 3–6; Figure 4C,D). Remarkably, the blockade of A_2A_R with SCH58261 (50 nM) prevented the inhibitory effect of theobromine on hippocampal LTP (13.31 ± 3.74% in the presence of SCH58261 vs. 51.22 ± 7.72% in the presence of SCH58261 + theobromine; *p* < 0.05, n = 3–6), an increase probably related to the disrupted interaction of A_2A_R and A_1_R to reveal and A_1_R-medated effect of theobromine upon blockade of A_2A_R [42,43].

To further confirm the involvement of A_2A_R in the effect of theobromine on hippocampal LTP, we tested the effects of theobromine in hippocampal slices from mice where A_2A_R were selectively deleted, either from the whole animal (gA_2A_R-KO mice) or only from forebrain neurons (fbA_2A_R-KO mice). In hippocampal slices from both mouse strains, the enhancement of basal synaptic transmission by 30 μM theobromine was maintained (37.93 ± 5.80% for gA_2A_R-KO and 43.59 ± 9.27% for fbA_2A_R-KO vs. baseline; *p* < 0.001, n = 12 and *p* < 0.01, n = 5, respectively; Figure 5A,B,E,F). However, theobromine was unable to significantly alter LTP magnitude in slices from gA_2A_R-KO mice (65.75 ± 9.59% in the absence vs. 80.66 ± 6.17% in the presence of theobromine; *p* > 0.05, n = 12; Figure 5C,D) and from fbA_2A_R-KO mice (50.69 ± 10.95% in theobromine-treated slices vs. 55.40 ± 9.72% in control slices; *p* > 0.05, n = 5; Figure 5G,H).

### 2.5. Theobromine and Caffeine Abrogate the Impairment of Synaptic Plasticity and Neuronal Viability Triggered by β-Amyloid Peptides

Synaptic dysfunction is at the core of initial changes in early AD [44,45]. Modelling early AD, a two-week exposure to oligomeric Aβ_1–42_ significantly impairs synaptic plasticity, typified by a decrease in LTP magnitude [20]. This is mimicked by the acute exposure of mouse hippocampal slices to oligomeric Aβ_1–42_ (50 nM during 40 min) (unpublished preliminary data), which replicates the decrease in LTP magnitude observed in the Aβ-injection model [20]. We now report that a previous superfusion of mouse hippocampal slices with theobromine (30 μM) prevented the noxious impact of Aβ_1–42_ on hippocampal LTP magnitude (25.50 ± 5.05% without vs. 53.88 ± 5.58% with theobromine; *p* < 0.01, n = 5–6; Figure 6A,B). Remarkably, the exposure to caffeine (50 μM) also prevented the noxious impact of Aβ_1–42_ on hippocampal LTP magnitude (13.40 ± 2.61% without vs. 44.12 ± 6.07% with caffeine; *p* < 0.001, n = 7; Figure 6C,D).

Aβ peptides were previously demonstrated to have a neurotoxic effect, triggering an apoptotic response in neuronal cultures, which was characterized with different methods [19,46]. Accordingly, the exposure of primary cultures of mouse cortical neurons to Aβ_1–42_ (500 nM during 24 h) triggered the appearance of 31.58 ± 4.25% (n = 4) of apoptotic-like cells, displaying condensed and fragmented nuclei stained more intensely with DAPI (Figure 6E). This noxious effect was prevented by theobromine (30 μM) (13.46 ± 3.52% of apoptotic cells; *p* < 0.05, n = 4; Figure 6E,F) as well as by 50 μM caffeine (13.13 ± 2.81% of apoptotic cells; *p* < 0.05, n = 4; Figure 6E,G). Overall, this similar prevention of Aβ-induced synaptic deficits and neuronal toxicity by theobromine and caffeine indicates an analogous mechanism operated by both xanthines to afford a prophylactic neuroprotection against early AD.

## 3. Discussion

The present study clarifies how theobromine, a caffeine derivative and the major psychoactive component of cocoa, controls synaptic function in concentrations close to those monitored in the blood and brain parenchyma [39]. This involves the modulation of adenosine A_1_ and A_2A_ receptors in a manner similar to the effects of caffeine. The data also show that theobromine and caffeine can equally prevent the deficits of synaptic plasticity and of neuronal viability, induced by β-amyloid peptides that are purported culprits of AD.

Although it has been proposed that theobromine might affect different molecular targets [37,38,39,47], we now observed that the effects of theobromine on synaptic transmission and plasticity were completely occluded upon removal of endogenous extracellular adenosine, a neuromodulator known to mainly operate A_1_R and A_2A_R in the brain [48]. Indeed, the pharmacological blockade with DPCPX of A_1_R, which inhibits excitatory synaptic transmission in central glutamatergic synapses, namely in Schaffer fiber-CA1 pyramid synapses [49], fully prevents the effect of theobromine on basal synaptic transmission. This points to the conclusion that theobromine acts via A_1_R to control basal synaptic transmission. Additionally, the conversion of adenosine to inosine by ADA, which abolishes the excitatory effect of tonic A_2A_R activation in Schaffer fiber-CA1 pyramid synapses [40], abrogated the effect of theobromine on LTP. This effect is mimicked by the pharmacological inhibition of A_2A_R with the selective antagonist SCH58261, as well as by the genetic elimination of A_2A_R. This shows that theobromine acts via neuronal A_2A_R to modulate synaptic plasticity.

The present conclusion that theobromine seems to act solely through A_1_R and A_2A_R to control information flow in neuronal circuits is rather surprising, since the reported affinities of theobromine for adenosine receptors are in the high micromolar range [32,33,34,50]; in contrast, the impact of theobromine on hippocampal synaptic transmission and plasticity is strictly dependent on adenosine receptors, and these effects of theobromine are superimposable to these previously reported for caffeine [11]. This may partially result from particular binding kinetics to adenosine receptors of these two xanthines, a factor recently highlighted to be of crucial importance to understand the efficiency of drugs to activate adenosine receptors [51]. This similar efficiency of theobromine and caffeine to modulate synaptic function through adenosine receptors justifies the observations that the acute exposure to theobromine has discrete central stimulant effects [52] and mostly affects peripheral physiology with lower psychomotor properties, compared to caffeine [53,54], although the effects of theobromine are longer lasting, due to its longer half-life [55].

Notably, the regular consumption of theobromine has neuroprotective action, revealed both in animal and human studies (reviewed in [56,57,58]). Indeed, recent studies indicate that chocolate consumption associates with a lower risk of developing age-related cognitive decline [28]; also, there is a positive correlation between theobromine and Aβ_1–42_ levels in the plasma of AD patients [23] and that theobromine blocks the increase in Aβ levels induced by lard-enriched diet in rodents [30]. We now report that theobromine is equi-effective with caffeine to prevent Aβ-induced neurotoxicity, as well as Aβ-induced impairment of synaptic plasticity, one of the earliest alterations found in AD [59]. Since previous studies have identified that the synaptoprotective role of caffeine is mediated by A_2A_R (e.g., [9,60,61]) and we now observed that theobromine controls hippocampal synaptic plasticity through the antagonism of A_2A_R, it is reasonable to assume that theobromine prevents Aβ-induced dysfunction through the antagonism of A_2A_R. Notably, the previously proposed involvement of the CaMKII/CREB/BDNF pathway in the effects of theobromine on memory performance [31,62] is compatible with the involvement of A_2A_R that are well known to control this pathway (reviewed in [15]). This understanding of how theobromine impacts on synaptic plasticity is relevant to mechanistically understand the benefits provided by chocolate and other cocoa-based substances for memory and cognitive function upon ageing and brain disorders, such as AD [28,58,63]. Furthermore, the near equi-effective neuroprotection afforded by caffeine and theobromine is supportive of the long-lasting beneficial effects, resulting from the consumption of caffeinated beverages, namely of caffeinated coffee, which may involve an initial protection afforded by caffeine that is prolonged by the protective effect of caffeine metabolites, such as theobromine, which has a longer half-life [55]. Moreover, although caffeine-mediated neuroprotection seems to be mostly accounted by the control of neuronal A_2A_R, it is possible that the neuroprotective effects of methylxanthines might also involve glia (astrocytes and/or microglia)-mediated effects, which remains to be adequately characterized.

One question that remains unsolved is the toxicity of theobromine in most animals (e.g., [26,27]), which is not present in humans. Since both theobromine and caffeine, which has a limited toxicity, can antagonize adenosine receptors, it is unlikely that adenosine receptors are involved in chocolate toxicosis in animals, which might instead involve other molecular targets, such as poly(ADP-ribose)polymerase-1 [37] or sirtuin-1 [37,38,39,45]. These theobromine-operated molecular targets involved in chocolate toxicosis are likely to have a greater peripheral rather than central impact, since we now observed that the exposure of neurons to theobromine did not cause a direct neurotoxicity. However, further studies are required to directly identify the targets operated by theobromine to trigger toxicosis in animals.

In conclusion, our results demonstrate that theobromine, analogously to caffeine, can alter synaptic functioning through the control of adenosine receptors, modulating synaptic transmission via A_1_R and synaptic plasticity by acting as an A_2A_R antagonist. Furthermore, theobromine is equi-effective with caffeine to prevent the impairment of synaptic plasticity and the neuronal loss in conditions designed to replicate what occurs in early AD, which provides a new mechanistic insight on the prophylactic benefits, resulting from the consumption of chocolate and other cocoa-derived products.

## 4. Methods and Materials

### 4.1. Ethical Approval

Animal procedures were performed in accordance with the guidelines of the European community guidelines (EU Directive 2010/63/EU) and the Portuguese law on animal care (1005/92) and approved by the Ethical Committee of the Center for Neuroscience and Cell Biology of Coimbra.

All efforts were made to reduce the number of animals used and to minimize their stress and discomfort. Thus, the animals were anesthetized in halothane atmosphere before decapitation, and, whereas the hippocampus was used in this study, other tissues from these animals were collected for use in different projects at our research center.

### 4.2. Animals

We used C57bl\6j mice with 8–10 weeks of age, obtained from Charles River (Barcelona, Spain), global A_2A_R knockout mice (gA_2A_RKO), and forebrain A_2A_R knockout mice (FbA_2A_RKO) with the same age and genetic background, which were generated in our colony, as previously described [64,65]. The animals were housed under controlled temperature (23 ± 2 °C), subject to a fixed 12 h light/dark cycle, with free access to food and water.

Pregnant female C57bl\6j mice for embryo collection were obtained from parallel breeding at CNC and maintained on a 12 h light/dark cycle with food and water ad libitum. The mating procedure was limited to 12 h, and the day of vaginal plug observation was considered embryonic day 0.5 (E0.5).

### 4.3. Drugs

Theobromine, caffeine, adenosine deaminase (ADA), and 8-cyclopentyl-1,3-dipropylxanthine (DPCPX) were from Sigma (St. Louis, MI, USA). SCH58261 was obtained from Tocris (Bristol, UK). The Aβ_1–42_ peptide was purchased from Bachem (Bubendorf, Germany).

Theobromine was freshly prepared in milliQ water to a stock concentration of 1 mM. Caffeine was prepared in milliQ H_2_O to a stock concentration of 100 mM. Stock solutions of DPCPX (5 mM) and SCH58261 (5 mM) were prepared in dimethylsulfoxide (Sigma) and dilutions were prepared in ACSF or Krebs solution, controlling for the impact of the residual amount of dimethylsulfoxide. ADA, DPCPX, and SCH58261 were used in supramaximal but selective concentrations, respectively, 2 U/mL [66], 100 nM [67], and 50 nM [68]. Aβ_1–42_ was dissolved in water to obtain a solution mostly composed of Aβ low molecular weight oligomers [19,69].

All other chemical substances used, unless stated otherwise, were from Sigma (St. Louis, MI, USA).

### 4.4. Extracellular Electrophysiological Recordings

Following decapitation, the mouse brain was quickly removed and placed in ice-cold, oxygenated (95% O_2_, 5% CO_2_) artificial cerebrospinal fluid (ACSF; in mM: 124.0 NaCl, 4.4 KCl, 1.0 Na_2_HPO_4_, 25.0 NaHCO_3_, 2.0 CaCl_2_, 1.0 MgCl_2_, 10.0 glucose). Using a McIlwain tissue chopper (Brinkmann Instruments, Long Island, NY, USA), slices (400 µm-thick) from the dorsal hippocampus were cut transverse to the long axis of the hippocampus and placed in a holding chamber with oxygenated ACSF at room temperature (RT). Slices were allowed to recover for at least 1 h prior to recording, when they were transferred to a submerged recording chamber and superfused at 3 mL/min with oxygenated ACSF kept at 30.5 °C.

The configuration of the extracellular recordings was as previously described [8,29] with the stimulating bipolar concentric electrode placed in the proximal CA1 stratum radiatum for stimulation of the Schaffer fibers and the recording electrode, filled with 4 M NaCl (2–5 MΩ resistance), was placed in the CA1 stratum radiatum, targeting the distal dendrites of pyramidal neurons. Stimulation was performed using either a Grass S44 or a Grass S48 square pulse stimulator (Grass Technologies, West Warwick, RI, USA) and every 20 s with rectangular pulses of 0.1 ms. After amplification (ISO-80, World Precision Instruments, Hitchin, Hertfordshire, UK), the recordings were digitized (BNC-2110, National Instruments, Newbury, UK), averaged in groups of 3, and analyzed using the WinLTP version 2.10 software (WinLTP Ltd., Bristol, UK) [70]. The intensity of stimulation was chosen between 50–60% of maximal fEPSP response, determined on the basis of input/output curves, in which the percentage of maximum fEPSP slope was plotted versus stimulus intensity.

Alterations of synaptic transmission were quantified as the % modification of the average value of the fEPSP slope taken from 25 to 30 min after beginning theobromine application, depending on the experiment or from 10 to 15 min, after beginning the application of modifying drugs (adenosine deaminase, DPCPX or SCH58261), in relation to the average value of the fEPSP slope during the 5 min that preceded application of each modifying drug. To quantify the effect of theobromine applied after other modifying drugs, we compared fEPSP slope values in the 5 min that preceded application of theobromine (baseline) in relation to the average values from 25 to 30 min after theobromine exposure.

Long-term potentiation (LTP) was induced by a high-frequency stimulation train (100 Hz for 1 s). LTP was quantified as the percentage change between two values: the average slope of the 10 averaged potentials taken between 50 and 60 min, after LTP induction, in relation to the average slope of the fEPSP, measured during the 10 min that preceded LTP induction. The effect of drugs on LTP was assessed by comparing LTP magnitude in the absence and presence of the drug in experiments carried out in different slices from the same animal.

### 4.5. Cell Culture

Primary cultures of cortical neurons were prepared as previously described [71]. Briefly, cortices from C57BL/6 E17.5 mouse embryos were dissected and digested for 15 min at 37 °C with trypsin (0.125%; type II-S, from porcine pancreas; Sigma) and DNAse (50 μg/mL; DNAse I from bovine pancreas; Sigma) in calcium- and magnesium-free sterile Hank’s balanced salt solution (137 mM NaCl, 5.36 mM KCl, 0.44 mM KH_2_PO_4_, 4.16 mM NaHCO_3_, 0.34 mM Na_2_HPO_4_, 5 mM glucose, 5 mM sodium pyruvate, 10 mM HEPES, pH 7.2). An equal amount of DMEM (Sigma) supplemented with 0.44 mM NaHCO_3_, 5 mM sodium pyruvate, 10% fetal bovine serum (Thermo Fisher Scientific, Waltham, MA, USA), and 1% penicillin/streptomycin (Thermo Fisher Scientific), pH 7.3, was added, and cells were mechanically dissociated, centrifuged at 170× *g* for 5 min at RT, and resuspended in fresh DMEM, prior to seeding at a density of 2.0 × 10^4^ cells/cm^2^ on poly-D-lysine- (100 μg/mL; Sigma) coated coverslips. After 2 h, the medium was replaced by Neurobasal medium (Thermo Fisher Scientific) with 2% B27 supplement (Thermo Fisher Scientific), 50 U/mL penicillin, 50 μg/mL streptomycin (Thermo Fisher Scientific), and 2 mM glutamine (GlutaMAX-I, Gibco, Life Technologies, Carlsbad, CA, USA), and cells were grown at 37 °C in an atmosphere of 95%/5% air/CO_2_ until they were used 5 days later.

### 4.6. Peptide Treatment and Immunocytochemistry

Based on our previous characterization of Aβ_1–42_-induced neurotoxicity [19], cultured cortical neurons in glass coverslips were treated with oligomeric Aβ_1–42_ (500 nM) for 24 h in the presence/absence of either theobromine (30 μM) or caffeine (50 μM). The drugs were added into the culture medium at the 5th culturing day. After a 24 h exposure period, cells were washed with PBS and fixed with a 4% paraformaldehyde solution (pH 7.4), for 30 min at RT. The cells were permeabilized with 0.2% Triton X-100/PBS for 10 min at RT and blocked with 3% bovine serum albumin and 5% horse serum for 1 h before incubation with 4′,6-diamidino-2-phenylindole (DAPI) (1:5000). After being washed in PBS to remove the excess staining, the cells were mounted with the DakoCytomation fluorescent medium and visualized in a fluorescence microscope. Apoptotic-like neurons were identified as cells with condensed nuclei with an irregular form, often fragmented, displaying a more intense light blue staining with DAPI. The counting of apoptotic-like neurons versus the total number of neurons was calculated in 4 fields per coverslip.

### 4.7. Statistics

The values presented are mean ± S.E.M. with the number of determinations (n, i.e., slices or cells from different mice) in each experimental condition, indicated in the legend to the figures. The comparison of two experimental conditions was performed using either a paired or unpaired Student’s *t* test. Otherwise, statistical analysis was performed by a two-way analysis of variance (ANOVA), followed by a Tukey’s post hoc test. *p* < 0.05 was considered to represent statistical significance. Statistical analysis was performed using GraphPad Prism software (GraphPad Software, San Diego, CA, USA).

## Figures and Tables

**Figure 1 ijms-23-10510-f001:**
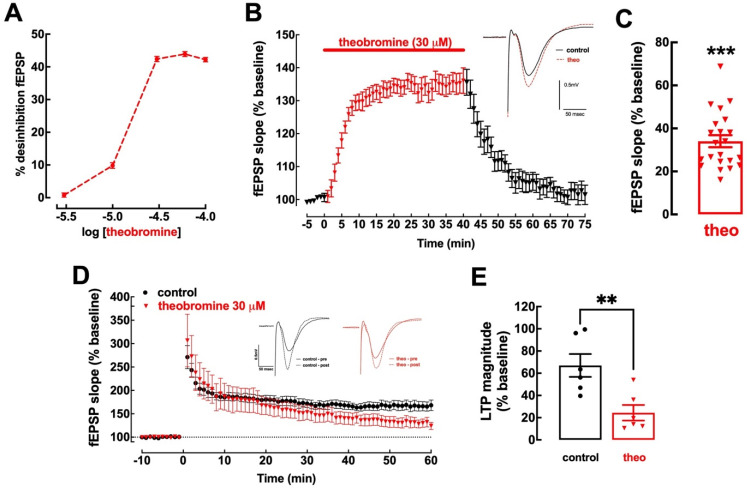
Theobromine alters basal synaptic transmission and synaptic plasticity. Theobromine increased basal synaptic transmission in Schaffer fibers-CA1 pyramid synapses of mouse hippocampal slices in a concentration-dependent manner, reaching a plateau after 30 μM (**A**). Theobromine (theo, 30 μM) rapidly and reversibly increased basal synaptic transmission (**B**,**C**), and the insert in (**B**) displays representative field extracellular postsynaptic potential (fEPSP) recorded in ACSF medium (control conditions, black filled line) and in the presence of theobromine (red dashed line). Additionally, theobromine significantly decreased the magnitude of long-term potentiation (LTP): the time course of fEPSP recordings (**D**) shows a sustained increase in fEPSP slope after delivering a high-frequency stimulation train (HFS: one train of 100 pulses of 1 Hz for 1 s), which is larger in ACSF (black symbols) than in the presence of 30 μM theobromine (red symbols), as quantified in (**E**). The inserts in (**D**) show two superimposed pairs fEPSP collected before (filled traces) and 60 min after HFS (dashed lines) in control conditions (black traces on the left) and in the presence of theobromine (red traces on the right). Data are mean ± S.E.M. of 5–22 (**A**), 22 (**B**,**C**) or 6 (**D**,**E**) experiments. The asterisks denote significant differences: ** *p* < 0.01 vs. control, Student’s *t* test (**E**); *** *p* < 0.001 vs. baseline, one-sample *t* test (**C**).

**Figure 2 ijms-23-10510-f002:**
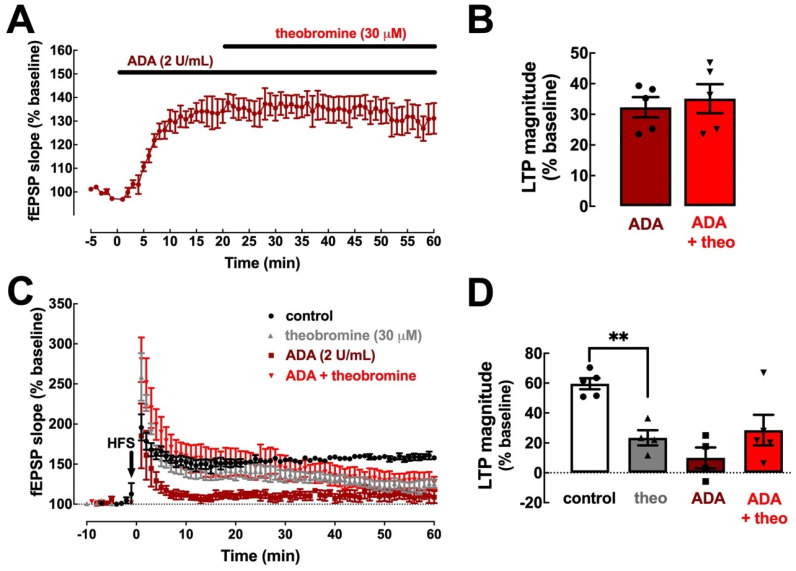
Clearance of endogenous extracellular adenosine abolishes the effects of theobromine on synaptic transmission and plasticity. Adenosine deaminase (ADA, 2 U/mL) enhanced hippocampal basal synaptic transmission (**A**,**B**) and abrogates the effects of theobromine (30 μM). ADA (2 U/mL) also decreased the magnitude of long-term potentiation (**C**,**D**) induced by a high-frequency stimulation train (HFS: one train of 100 pulses of 1 Hz for 1 s) in Schaffer fiber-CA1 pyramid synapses of mouse hippocampal slices and abrogates the effects of theobromine (theo, 30 μM). This shows that these measured effects of theobromine are strictly dependent on the presence of endogenous extracellular adenosine. Data are mean ± S.E.M. of 5 (**A**,**B**) and 4–5 (**C**,**D**) experiments. The asterisks denote significant differences: ** *p* < 0.01 vs. control, two-way ANOVA with Tukey’s post hoc test.

**Figure 3 ijms-23-10510-f003:**
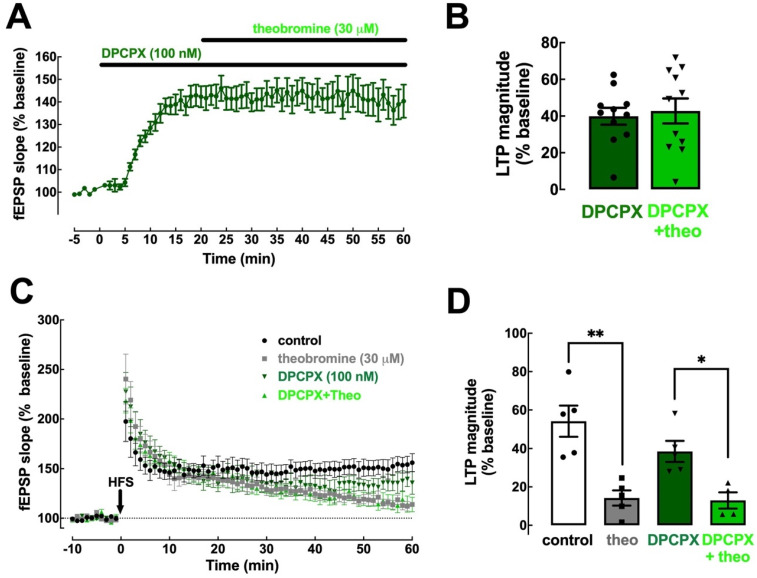
Blockade of A_1_R eliminates the impact of theobromine on basal synaptic transmission but not on synaptic plasticity. The selective A_1_R antagonist DPCPX (100 nM) enhanced basal synaptic transmission and occluded the effect of theobromine (theo, 30 μM) on synaptic transmission (**A**,**B**). In contrast, DPCPX did not modify the magnitude of long-term potentiation (LTP), induced with a high-frequency stimulation train (HFS: one train of 100 pulses of 1 Hz for 1 s), recorded in Schaffer fiber-CA1 pyramid synapses of mouse hippocampal slices and failed to modify the effects of theobromine on LTP magnitude (**C**,**D**). Data are mean ± S.E.M. of 11 (**A**,**B**) and 5–6 (**C**,**D**) experiments. The asterisks denote significant differences: * *p* < 0.05 and ** *p* < 0.01, two-way ANOVA with Tukey’s post hoc test.

**Figure 4 ijms-23-10510-f004:**
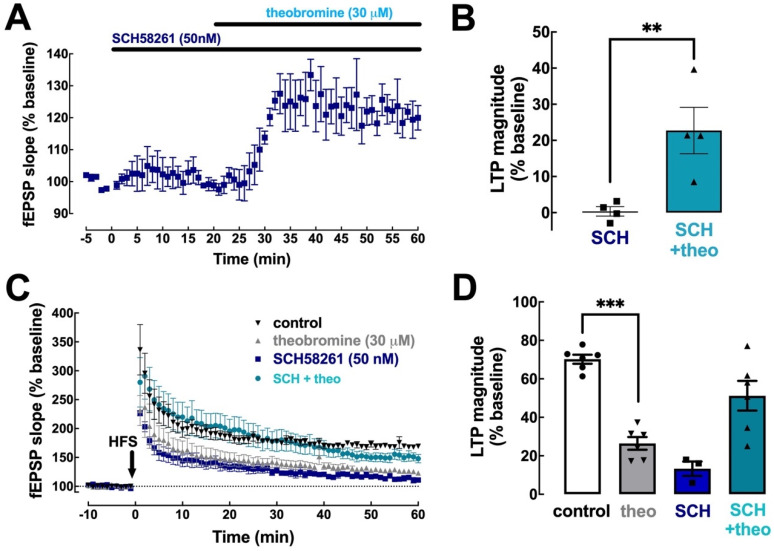
Blockade of A_2A_R prevents the effect of theobromine on long-term potentiation. The selective A_2A_R antagonist SCH58261 (SCH, 50 nM) did not modify basal synaptic transmission nor altered the effect of theobromine (theo, 30 μM) on basal synaptic transmission in mouse hippocampal slices (**A**,**B**). In the presence of SCH58261 (50 nM), theobromine (30 μM) failed to decrease LTP magnitude (**C**,**D**), induced with a high-frequency stimulation train (HFS: one train of 100 pulses of 1 Hz for 1 s) in Schaffer fiber-CA1 pyramid synapses of mouse hippocampal slices. Data are mean ± S.E.M. of 4 (**A**,**B**) and 3–6 (**C**,**D**) experiments. The asterisks denote significant differences: ** *p* < 0.05 vs. SCH58261, Student’s *t* test (**B**); *** *p* < 0.001 vs. control, two-way ANOVA with Tukey’s post hoc test.

**Figure 5 ijms-23-10510-f005:**
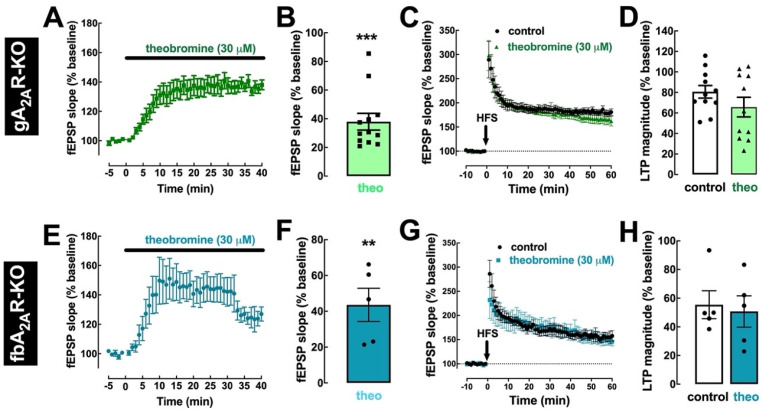
Genetic deletion of A_2A_R prevents the effect of theobromine on long-term potentiation. In slices from knockout mice where A_2A_R were deleted globally (gA_2A_R-KO, (**A**–**D**)) or selectively in forebrain neurons (fbA_2A_R-KO, (**E**–**H**)), theobromine (theo, 30 μM) still enhanced basal synaptic transmission (**A**,**B**,**E**,**F**), but did not modify the magnitude of LTP (**C**,**D**,**G**,**H**), induced with a high-frequency stimulation train (HFS: one train of 100 pulses of 1 Hz for 1 s) in Schaffer fiber-CA1 pyramid synapses of mouse hippocampal slices. Data are mean ± S.E.M. of 12 (**A**,**B**), 11 (**C**,**D**), and 5 (**E**–**H**) experiments. The asterisks denote significant differences: ** *p* < 0.01, *** *p* < 0.001 vs. baseline, one-sample *t* test.

**Figure 6 ijms-23-10510-f006:**
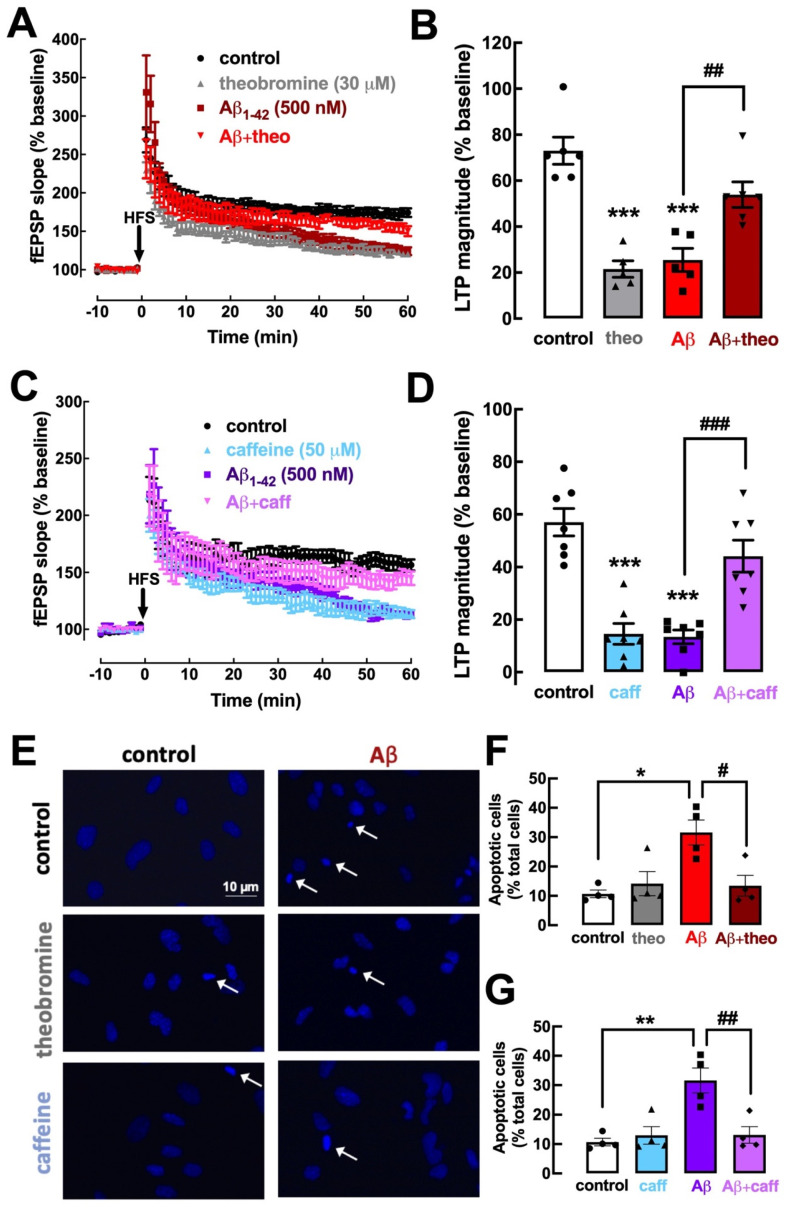
Theobromine and caffeine prevent synaptic plasticity deficits and neurotoxicity induced by Aβ_1–42_ modelling early Alzheimer’s disease. The superfusion of the mouse hippocampal slices with Aβ_1–42_ (50 nM) during 40 min induced a significant decrease in LTP magnitude (**A**–**D**) that was prevented by the presence of either theobromine (theo, 30 μM) (**A**,**B**) or caffeine (caff, 50 μM) (**C**,**D**). Likewise, primary neuronal cultures treated with Aβ_1–42_ (500 nM) during 24 h exhibit a significant increase in the number of apoptotic neurons (**E**–**G**), which were identified as cells with condensed nuclei with an irregular form often fragmented, displaying a more intense light blue staining with DAPI, as indicated by the arrows in the photographs (**E**). This neurotoxicity of Aβ_1–42_ was prevented both by theobromine (30 μM) (**E**,**G**) or caffeine (50 μM) (**F**,**G**). Data are mean ± S.E.M. of 5–6 (**A**,**B**), 7 (**C**,**D**), and 4 (**E**–**G**) experiments, where the counting of apoptotic-like neurons versus the total number of neurons was calculated in 4 field per coverslip. The scale bar (10 μm) of the first photograph in (**E**) applies to all other photographs. The asterisks denote significant differences: * *p* < 0.05, ** *p* < 0.01, *** *p* < 0.001 vs. control, two-way ANOVA with Tukey’s post hoc test; # *p* < 0.05, ## *p* < 0.01, ### *p* < 0.001 vs. Aβ_1–42_, two-way ANOVA with Tukey’s post hoc test.

## Data Availability

The data that support the findings of this study are available from the corresponding author upon reasonable request.

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
