# Peer review of "Thebromine Targets Adenosine Receptors to Control Hippocampal Neuronal Function and Damage"

_ijms, 2022, doi:10.3390/ijms231810510_

Round 1
Reviewer 1 Report
The MS the THEBROMINE and its effect on Adenosine receptor is really interesting and complete.
Minor
I would suggest to improve the introduction which is quite superficial by adding context and stating why this work is important.
could you also write a clear hypothesis and objective in introduction
Figure 1, 2, 3 and 4 B is a control missing ?
Figure 6E what is the staining used, it's confusing because you show 2 different things in blue or it is just DAPI (Materials and methods)? If it's just a DAPI staining as the materials and methods section suggests, how can you make a difference between dying and living cells? If you only stained with DAPI i would highly suggest to counterstain with another maker.
Could you add the scale bars in the picture and the length of it in the legend.
In the legend: "during 24 exhibit a" I think it misses something after "24".
In the discussion you mentioned some limitations could you elaborate?
it would be interesting to discuss if in an animal or human THEBROMINE would have a direct impact on neurons or also via microglia which can support neurons.
Author Response
Reviewer 1
The MS the THEBROMINE and its effect on Adenosine receptor is really interesting and complete.
Minor
I would suggest to improve the introduction which is quite superficial by adding context and stating why this work is important.
could you also write a clear hypothesis and objective in introduction
We fully agree with the Reviewer that the Introduction lacked a clear statement of our goal and working hypothesis, which is now introduced.
Figure 1, 2, 3 and 4 B is a control missing ?
We carefully checked that all controls are displayed in the Figures. As described in the legends to the Figures, the effect of theobromine on basal synaptic transmission is calculated versus baseline (0%), whereas the effect of theobromine on LTP amplitude is calculated by comparison of LTP amplitude in the absence versus in the presence of theobromine.
Figure 6E what is the staining used, it's confusing because you show 2 different things in blue or it is just DAPI (Materials and methods)? If it's just a DAPI staining as the materials and methods section suggests, how can you make a difference between dying and living cells? If you only stained with DAPI i would highly suggest to counterstain with another maker.
We do apologize for our lack of clarity in explaining the methodology used to estimate amyloid peptide-induced neurotoxicity. We now describe in the Methods’ section, as well as in the Legend to Figure 6, that apoptotic-like neurons were identified as cells with condensed nuclei with an irregular form often fragmented, displaying a more intense light blue staining with DAPI.
Could you add the scale bars in the picture and the length of it in the legend.
Corrected as suggested.
In the legend: "during 24 exhibit a" I think it misses something after "24".
This was now corrected. Thanks very much for noting this mistake.
In the discussion you mentioned some limitations could you elaborate?
As recommended by the Reviewer, we now specify some putative alternative targets operated by theobromine to trigger toxicosis in animals and we directly state the need to design additional studies to clarify this issue.
it would be interesting to discuss if in an animal or human THEBROMINE would have a direct impact on neurons or also via microglia which can support neurons.
We now inserted in the Discussion of the revised manuscript a sentence to address the future need to explore if and how theobromine-mediated actions on glia cells might contribute for neuroprotection.
Reviewer 2 Report
The manuscript by Valada and co-authors investigated the molecular mechanism of the theobromine. They demonstrate that theobromine, analogously to caffeine, might alter synaptic functioning modulating synaptic transmission through the control of adenosine receptors A1R and synaptic plasticity by acting as an antagonist of the adenosine receptors, A2AR. Additionally, they showed that theobromine, as caffeine, is able to prevent the impairment of synaptic plasticity and neuronal loss in early AD.
The manuscript is interesting and well-designed. Results are clearly presented and discussed.
I have only a few suggestions.
Major pints.
The authors have to include in the method section the procedure for evaluating and counting apoptotic cells.
The results of apoptotic cells must be improved. I suggest conducting the experiments with a more appropriate method.
Minor points
The quality of Figure 1 must be improved to facilitate readers. I suggest the use of colored lines.
Author Response
The manuscript by Valada and co-authors investigated the molecular mechanism of the theobromine. They demonstrate that theobromine, analogously to caffeine, might alter synaptic functioning modulating synaptic transmission through the control of adenosine receptors A1R and synaptic plasticity by acting as an antagonist of the adenosine receptors, A2AR. Additionally, they showed that theobromine, as caffeine, is able to prevent the impairment of synaptic plasticity and neuronal loss in early AD.
The manuscript is interesting and well-designed. Results are clearly presented and discussed.
I have only a few suggestions.
Major pints.
The authors have to include in the method section the procedure for evaluating and counting apoptotic cells.
We very much thank the Reviewer for noting that we did not describe the quantification of apoptotic-like neurons, which we now inserted in the Methods’ section of the revised manuscript.
The results of apoptotic cells must be improved. I suggest conducting the experiments with a more appropriate method.
Although we agree with the Reviewer that different methods are available to evaluate apoptotic-like features in neurons, our previous experience with different such quantification methods documented in several previous publications (e.g. Rebola et al., 2005, Neurochem Int 47:317-25; Silva et al., 2007, Neurobiol Dis 27:182-9; Canas et al., 2009, J Neurosci 29:14741-51) has shown us that the use of the inspection of the pattern of DAPI staining is sufficiently reliable to estimate what we aim to do in the present study, that is, to estimate the neurotoxicity upon exposure to amyloid peptides. In fact, our present goal is not to define mechanisms of amyloid peptides-induced neurotoxicity, but only to compare the neuroprotective ability of two methylxanthines with the previously well-established neuroprotection afforded by selective A2A receptor antagonists in neurons exposure to amyloid peptides. To cope with the Reviewer’s concern, we now explicitly stated in the description of the results that Aβ-induced apoptotic-like neurotoxicity in cultured neurons has previously been identified and that we now used the pattern of DAPI staining as a proxy method to identify these apoptotic-like neurons upon exposure to Aβ.
Minor points
The quality of Figure 1 must be improved to facilitate readers. I suggest the use of colored lines
As suggested, we now inserted a coloured line in all panels of Figure 1.
Round 2
Reviewer 2 Report
no comments